# mTOR-Dependent Autophagy Regulates Slit Diaphragm Density in Podocyte-like *Drosophila* Nephrocytes

**DOI:** 10.3390/cells11132103

**Published:** 2022-07-02

**Authors:** Dominik Spitz, Maria Comas, Lea Gerstner, Séverine Kayser, Martin Helmstädter, Gerd Walz, Tobias Hermle

**Affiliations:** 1Renal Division, Department of Medicine, Faculty of Medicine and Medical Center, University of Freiburg, 79106 Freiburg, Germany; dominik.spitz@uniklinik-freiburg.de (D.S.); lea.gerstner@uniklinik-freiburg.de (L.G.); severine.kayser@uniklinik-freiburg.de (S.K.); martin.helmstaedter@uniklinik-freiburg.de (M.H.); gerd.walz@uniklinik-freiburg.de (G.W.); 2CIBSS—Centre for Integrative Biological Signalling Studies, 79106 Freiburg, Germany

**Keywords:** nephrocyte, *Drosophila*, podocyte, mTOR, autophagy, slit diaphragm, nephrin

## Abstract

Both mTOR signaling and autophagy are important modulators of podocyte homeostasis, regeneration, and aging and have been implicated in glomerular diseases. However, the mechanistic role of these pathways for the glomerular filtration barrier remains poorly understood. We used *Drosophila* nephrocytes as an established podocyte model and found that inhibition of mTOR signaling resulted in increased spacing between slit diaphragms. Gain-of-function of mTOR signaling did not affect spacing, suggesting that additional cues limit the maximal slit diaphragm density. Interestingly, both activation and inhibition of mTOR signaling led to decreased nephrocyte function, indicating that a fine balance of signaling activity is needed for proper function. Furthermore, mTOR positively controlled cell size, survival, and the extent of the subcortical actin network. We also showed that basal autophagy in nephrocytes is required for survival and limits the expression of the *sns* (nephrin) but does not directly affect slit diaphragm formation or endocytic activity. However, using a genetic rescue approach, we demonstrated that excessive, mTOR-dependent autophagy is primarily responsible for slit diaphragm misspacing. In conclusion, we established this invertebrate podocyte model for mechanistic studies on the role of mTOR signaling and autophagy, and we discovered a direct mTOR/autophagy-dependent regulation of the slit diaphragm architecture.

## 1. Introduction

Autophagy is a fundamental catabolic process that contributes to cell homeostasis by degrading damaged or unnecessary proteins, organelles, or lipids [1]. The resulting macromolecules can be recycled and reused by the cell. Autophagy is negatively regulated by the mechanistic target of the rapamycin (mTOR) pathway [2]. The mTOR pathway is an evolutionary, highly conserved key regulator of cytoskeleton organization, cellular growth, proliferation, metabolism, and survival [3]. The mTOR protein is the central component of this signaling pathway, and it interacts with two different sets of proteins to form two multi-protein complexes, namely mTORC1 and mTORC2 [4]. An abundance of nutrients, amino acids, and growth factors, among others, will activate the mTOR pathway, which will, in turn, inactivate autophagy [5]. In contrast, starvation, endoplasmic reticulum (ER) stress, or depletion of growth factors will inhibit the mTOR pathway leading to upregulation of autophagy [4]. The recycled cellular components via autophagy will consequently support the reactivation of mTOR [4]. Well-functioning of autophagy is crucial for cell viability and function, intracellular quality control, and plays a role in a range of diseases, including renal disease [6,7].

The glomerular filtration barrier (GFB) consists of a fenestrated endothelium, the glomerular basement membrane, and the slit membrane, which is formed by the podocytes [8]. Podocytes are highly specialized epithelial cells that interact with each other through their foot processes with a unique intercellular junction called a slit diaphragm, and blood is filtrated through these slits [9]. Nephrin and NEPH1 are the main proteins forming the slit diaphragm that allows the passage of water and electrolytes but retains plasma components with a higher molecular weight, including most plasma proteins [10]. Podocyte dysfunction manifests with proteinuria and loss of renal function [11]. 

Both the mTOR and autophagy pathways maintain glomerular homeostasis, promote podocyte survival and function, slow down the progression of podocytopathies, and offer protection against glomerular injury and aging-related loss of function in the kidney [12,13,14,15,16,17,18]. Dysregulation of the mTOR and autophagy pathways leads to renal disease [19,20]. Interestingly, mTOR inhibitors regulate the expression of slit diaphragm proteins in podocytes [21,22,23,24]. In terms of autophagy, conditional proximal tubule-specific KO mice for Atg5 or Atg7 display enhanced renal injury compared to wild-type mice in response to induced acute kidney injury (AKI) or chronic kidney disease (CKD) [20]. Furthermore, it has been suggested that in diabetic CKD patients, the mTOR pathway is hyperactivated, resulting in the downregulation of autophagy and contributing to podocyte degeneration [25]. Podocyte-specific Atg5 knockout mice, which are autophagy-deficient, show an increase in proteinuria and ER stress [16] as well as accelerated diabetes-induced podocytopathy, glomerulosclerosis, and an increase in permeability of the GFB [26]. 

Pharmacological and genetic tools, primarily using in vitro and mouse models, have been used to understand the role of the mTOR and autophagy pathways in the renal system [19]. However, the current understanding of the mechanisms governing the roles of these pathways in the pathophysiology of renal disease is still very incomplete. *Drosophila* is a suitable model for the glomerular filtration barrier [27], harboring the garland cell nephrocytes, which are structurally and functionally similar to mammalian podocytes [28]. They form a slit diaphragm containing Sns and Kirre, the fly orthologs of the mammalian slit diaphragm proteins nephrin and NEPH1 [29,30]. Similarly, *Drosophila* has served as a model to study autophagy, particularly in salivary glands, fat tissue, midgut cells, ovaries, muscle, or the nervous system [31,32,33]. Many specific protocols and *Drosophila* lines are readily available to investigate autophagy [34,35,36]. However, very little work has been done on autophagy or the mTOR pathway in *Drosophila* nephrocytes. 

The current study investigated the effects of the mTOR pathway and autophagy using *Drosophila* garland cell nephrocytes. Activation and inhibition of the mTOR pathway show opposite effects on protein synthesis, cell size, survival, and actin cytoskeleton on nephrocytes. Furthermore, the activation and inhibition of mTOR differentially affects the nephrocyte ultrastructure, increasing and decreasing labyrinthine channels, respectively. This results in loss of function of nephrocytes. mTOR inhibition, but not activation, upregulating autophagy and disrupting the spacing of nephrocytes slit diaphragms. Importantly, this effect can be rescued by genetically blocking autophagy. Thus, our work demonstrates *Drosophila* nephrocytes as an attractive model to investigate the role of mTOR signaling and autophagy pathways in podocytopathies.

## 2. Materials and Methods

### 2.1. Fly Strains and Husbandry

Flies were reared on standard food at 25 °C. Overexpression and transgenic RNAi studies were performed using the *UAS/GAL4* system at 31 °C. Stocks obtained from the Bloomington Drosophila Stock Center (Bloomington, USA) were *UAS-Rheb* (#9688), *UAS-Tor-DN* (#7013), *UAS-Tor*-RNAi (#33951), *UAS-Tsc1*-RNAi (#31039), *UAS-Atg5*-RNAi (#34899), *UAS-Atg5*-RNAi.2 (#27551), and *UAS-GFP-mCherry-Atg8a* (#37749), and as controls, *UAS-lacZ* (#3955) and *UAS-EGFP*-RNAi (#41553). The stock *sns>EGFP^NLS^* was obtained from S. Abmayr [30]. To control expression in garland cell nephrocytes, we used *prospero-GAL4* (obtained from B. Denholm via M. Helmstädter) and *dorothy-GAL4* (#6903; BDSC).

### 2.2. Immunofluorescence Studies, TUNEL Detection and LIVE/DEAD Staining Using Drosophila Tissue

For immunofluorescence, nephrocytes were dissected and fixed for 20 min in phosphate-buffered saline (PBS) containing 4% paraformaldehyde. After fixation, staining was performed according to standard procedures. The following antibodies were used: guinea pig anti-Sns (1:100), mouse anti-Pyd (PYD2, DSHB, Iowa City, USA; 1:100), mouse anti-Rab7 (DSHB, Iowa City, USA; 1:100), Alexa Fluor^®^ 647 anti-HRP (#323-605-021, Jackson Immuno Research, Ely, UK; 1:200), and Phalloidin-iFluor 488 (#ab176753, abcam, Cambridge, UK; 1:200). TUNEL-positive cells were visualized using the In Situ Cell Death Detection Kit, Fluorescein (#11684795910, Roche, Mannheim, DE), according to the manufacturer’s instructions. Dead cells were identified using the LIVE/DEAD Fixable Orange (602) Viability Kit (#L34983, Thermo Fisher Scientific, Eugene, OR, USA) according to the manufacturer’s instructions. For imaging, a Zeiss LSM 880 laser scanning microscope was used by applying Airyscan technology for super-resolution microscopy. Image processing was done with the Zeiss ZEN Black and GIMP software and quantification was completed with ImageJ. Untreated *Drosophila* larvae, pupa, and adults were immobilized and imaged with the Leica S9i microscope (Leica Biosystems, Wetzlar, DE).

### 2.3. Channel Diffusion Assay

Nephrocytes were dissected and fixed for 5 min in PBS containing 4% paraformaldehyde. Cells were then incubated for 15 min in 0.2 mg/mL FITC-albumin (A9771, Sigma, St. Louis, MO, USA) to allow tracer diffusion into the channels. The procedure was completed after a second fixation step in paraformaldehyde with Hoechst 33342 (1:500) for 15 min. For imaging, a Zeiss LSM 880 laser scanning microscope (Carl Zeiss Microscopy, Jena, DE) with Airyscan was utilized. Quantification was done using ImageJ software.

### 2.4. Fluorescent Tracer Uptake

Fluorescent tracer uptake in nephrocytes was performed as previously described [28]. Briefly, nephrocytes were dissected in PBS and incubated with 200 µg/mL FITC-albumin (A9771, Sigma, St. Louis, MO, USA) for 30 s. After thorough washing in PBS, the nephrocytes were incubated for 10 min in Schneider’s medium (S0146, Sigma, St. Louis, MO, USA). After a fixation step of 5 min in 8% paraformaldehyde with Hoechst 33342 (1:500), cells were rinsed in PBS and mounted. For imaging, a Zeiss LSM 880 laser scanning microscope (Carl Zeiss Microscopy, Jena, DE) was used. Quantification was performed using ImageJ software.

### 2.5. Nile Red Staining of Neutral Lipids

Nephrocytes were dissected and fixed for 20 min in PBS containing 4% paraformaldehyde. Next, cells were incubated for 15 min in Nile Red (#72485, Sigma-Aldrich, St. Louis, MO, USA) diluted in dimethyl sulfoxide (100 µg/mL; D2650, Sigma, St. Louis, MO, USA). Cells were rinsed in PBS before mounting. Imaging was performed with a Zeiss LSM 880 laser scanning microscope (Carl Zeiss Microscopy, Jena, DE) using 488 and 561 nm wavelengths.

### 2.6. Electron Microscopy

For transmission electron microscopy (TEM), nephrocytes were dissected and fixed in 4% formaldehyde and 0.2% glutaraldehyde in 0.1 M cacodylate buffer, pH 7.4. TEM was carried out using standard techniques. The TEM imaging was done with a Talos L120C TEM (Thermo Scientific, Eugene, OR, USA).

### 2.7. Statistics

An unpaired *t*-test was used to determine the statistical significance between two interventions. A one-way ANOVA followed by Dunnett’s correction for multiple testing was used for multiple comparisons (GraphPad Prism software). Measurements were from distinct samples assuming Gaussian distribution. Asterisks indicate significance as follows: **p* < 0.05, ***p* < 0.01, ****p* < 0.001, *****p* < 0.0001. A statistically significant difference was defined as *p* < 0.05. Error bars indicate standard deviation (SD).

## 3. Results

### 3.1. mTOR Signaling Controls Spacing of Slit Diaphragms in Drosophila Nephrocytes

The mTOR signaling pathway integrates diverse cellular cues to regulate survival, the actin cytoskeleton, metabolism, and autophagy (Figure 1A). These pathways play a critical role in podocyte homeostasis and several podocytopathies [37,38], but the mechanistic role of the glomerular filtration barrier remains poorly understood. The nephrocytes of *Drosophila* function as storage kidneys and form a bi-layered filtration barrier, including molecular conserved slit diaphragms that control entry into the labyrinthine channels, a network of membrane invaginations (Figure 1B). In tangential sections, the slit diaphragm proteins Sticks and stones (Sns, the ortholog of nephrin) and Polychaetoid (Pyd, the ortholog of ZO-1) stain in a fingerprint-like linear pattern (Figure 1C–C’’) matching with dotted lines in cross-sections (insets Figure 1C–C’’). The *Drosophila* ortholog of *MTOR*, the target of rapamycin (*Tor*), is highly conserved in evolution (Appendix A). To explore the role of the conserved mTOR signaling pathway in the nephrocyte model, we specifically expressed an RNAi directed against *Tor* in these cells and observed wider spacing between the slit diaphragms (Figure 1D–D’’). To confirm the role of *Tor* loss-of-function independently, we expressed a dominant negative variant of *Tor* which resulted in an identical phenotype (Figure 1E–E’’). This implies a direct impact of this highly conserved signaling pathway on the filtration barrier in the *Drosophila* podocyte model. Distances between slit diaphragms remained unaltered upon activation of mTOR by silencing its inhibitor *Tsc1* or by overexpressing its activator *Rheb* (Figure 1F–G’’). The maximal density of slit diaphragms thus appears limited by other factors and cannot be enhanced by excessive mTOR signaling. Furthermore, quantitation of the distances between slit diaphragms confirmed significantly increased spacing upon Tor inhibition and lack of a significant effect after induction of mTOR signaling (Figure 1H). Finally, we visualized slit diaphragms directly using transmission electron microscopy (TEM) and verified these observations for *Tor*-RNAi and *UAS*-*Rheb* (Figure 1J–K, control Figure 1I).

### 3.2. Manipulation of mTOR Signaling Affects Nephrocyte Cell Size, Survival and sns (Nephrin) Expression

Next, we explored canonical functional aspects of mTOR signaling in nephrocytes. The mTOR pathway affects proteostasis, and mTOR inhibition is known to decrease nephrin expression [21,22,23,24], which might be the cause of the enlarged distance between slit diaphragms in nephrocytes with decreased Tor function. We examined the expression of the slit diaphragm protein Sns (nephrin) using a reporter line that employs a 2 kb enhancer sequence from intron 1 to control the expression of nuclear-targeted EGFP [30]. Quantitation of the reporter fluorescence indicated elevated *sns* expression after expression of *Tsc1*-RNAi, but no reduction of the expression of Sns (nephrin) after Tor inhibition (Figure 2A–D). Disinhibition of mTOR signaling thus promotes the synthesis of Sns (nephrin) proteins, but attenuation of mTOR signaling, conversely, does not decrease the activity of the *sns* promoter. Therefore, the reduced density of slit diaphragms associated with inhibition of Tor is not the consequence of reduced *sns* (nephrin) expression. To assess the impact of mTOR signaling on nephrocyte cell size, we measured the two-dimensional area bounded by the cell membrane in the equatorial plane after manipulation of mTOR. Nephrocytes were significantly smaller upon expression of *Tor*-RNAi or dominant negative *Tor* but enlarged by *Tsc1*-RNAi or overexpression of *Rheb* (Figure 2E–J). Consistent with the classical role of mTOR signaling and Tor-dependent podocyte hypertrophy [37], this pathway controls cell size in the podocyte-like nephrocytes, but cell size lacks an impact on slit diaphragm density. Since mTOR is involved in cell survival, we further analyzed cell death upon Tor manipulation using terminal deoxynucleotidyl transferase–mediated dUTP nick end-labeling (TUNEL) as a broad marker for cell death. Surprisingly, the vast majority of nephrocytes exhibited TUNEL-positive nuclei upon expression of *Tor*-RNAi or dominant negative *Tor,* while mTOR disinhibition using *Tsc1*-RNAi had no effect (Figure 2K–N, quantitation Figure 2O). This indicates that widespread initiation of cell death is the consequence of Tor inhibition and basal mTOR activity seems to promote cell survival in nephrocytes. Since TUNEL positivity may occur in nephrocytes that are still vital and even show functional hypertrophy [39], we utilized a live/dead detection method to determine the fraction of cells with immediately impaired viability after the expression of *Tor*-RNAi. This assay employs a lack of exclusion of a fixable viability dye resulting in elevated intracellular fluorescence intensity for the identification of dead cells. We observed a significant increase of dead cells upon mTOR inhibition, but only a minor fraction compared to the overall number of TUNEL-positive cells (Figure 2P–Q, quantitation Figure 2R). This supports the notion that Tor is required for nephrocyte survival and that, at least in third instar larvae, most cells are in the early stages of cell death while retaining an intact membrane barrier. Analysis in adults is precluded since expression of *Tor*-RNAi using *pros*-*GAL4* or *Dot*-*GAL4* both resulted in lethality within the pupal stage (Appendix A). This is likely due to expression in other tissues since nephrocytes appear dispensable for fly survival in conditions associated with husbandry in research [40]. mTOR inhibition further decreased larval and pupal body size (Appendix A).

### 3.3. mTOR Signaling Regulates the Labyrinthine Channel Network and Endocytic Function in Nephrocytes

Since mTOR signaling affects the organization of the actin cytoskeleton, we visualized actin filaments in nephrocytes using phalloidin. In control nephrocytes, a bright and dense line represented the cortical actin network while a more diffuse subcortical network and irregularly localized clusters were detectable (Figure 3A). In *Tor*-RNAi expressing nephrocytes, we observed a significant reduction of the subcortical actin, while conversely, subcortical actin was expanded upon expression of *Rheb* (Figure 3B–C, quantitation Figure 3D). Since the subcortical actin overlaps with the labyrinthine channel network, we visualized these membrane invaginations by passive diffusion of FITC-albumin after brief fixation. The channels were significantly shallower with Tor inhibition compared to control. Conversely, channels were significantly deeper and more expanded with the induction of mTOR signaling by *Tsc1*-RNAi and *UAS-Rheb* (Figure 3E–I, quantitation Figure 3J, Appendix A). We further assessed the uptake function of nephrocytes using endocytosis of FITC-albumin. This established assay reflects the nephrocyte’s role as an endocytic storage kidney [28]. To exclude a potential bias by channel depth, we modified the approach by introducing a brief post-exposure chase of 10 min after tracer exposition and washing steps. This serves to limit the detection to FITC-albumin that was subject to endocytosis and minimize any signal from tracer that resides within the channel after passage of the barrier but before endocytosis. Performing the functional assay in this manner, we observed a significant reduction of FITC-albumin endocytosis in nephrocytes for both mTOR loss- and gain-of-function (Figure 3K–M, quantitation Figure 3N). This suggests that increased channel depth by Tor activation does not translate to enhanced endocytosis, and a fine balance of Tor activity is required for optimal function of nephrocytes.

### 3.4. mTOR Inhibition Causes Elevated Autophagy in Nephrocytes

Inhibition of mTOR signaling disinhibits autophagy, and disruption of autophagy sensitizes podocytes for injury, eventually causing apoptosis [38]. We studied the occurrence of autophagocytic vesicles in nephrocytes using TEM. Autophagosomes are rare in control nephrocytes (Figure 4A) but became quite abundant in cells expressing *Tor*-RNAi, and vesicles representing various stages of autophagy were present (Figure 4B). The most abundant vesicles were the electron-dense autophagolysosomes, which is consistent with a strong expansion of Rab7-positive vesicles in *Tor*-RNAi or dominant negative *Tor* compared to control (Figure 4C–E). We performed Nile Red staining to rule out lipid droplets that also show high electron-density, but this lipid dye seemed much less abundant than autophagosomes (Appendix A). Thus, excessive autophagy in nephrocytes was confirmed upon Tor inhibition. To quantitatively study levels of autophagy in nephrocytes, we expressed a GFP-mCherry-Atg8a reporter in nephrocytes [41]. In the *Drosophila* fat body, this widely used reporter forms large, easily discernible vesicles as a read-out of autophagosomes [35]. In contrast, in nephrocytes, we observed a significant reduction of reporter-derived fluorescence upon mTOR inhibition (Figure 4F–G, quantitation Figure 4J). This suggests that the reporter protein, which is produced at a steady rate, is rapidly degraded within autophagolysosomes following elevated autophagy. This is compatible with the unusually high degradative capacity of nephrocytes, and excessive autophagy consumes most of the reporter protein. Conversely, direct inhibition of autophagy by expression of *Atg5*-RNAi and indirect inhibition of autophagy by excessive mTOR signaling (*Tsc1*-RNAi) significantly increased the reporter’s fluorescence (Figure 4H–I, quantitation Figure 4J). This confirms a plausible correlation between autophagy and reporter fluorescence intensity, which apparently accumulates in the absence of autophagy. The mCherry-derived fluorescence showed a more variable, irregular intensity (data not shown). This could be caused by this protein tag’s mild tendency to aggregate after cleavage while being more resistant to lysosomal degradation. However, GFP-derived fluorescence after the expression of *Atg5*-RNAi indicates basal autophagy in nephrocytes, and reporter fluorescence thus provides a convenient, quantifiable read-out for autophagy in these cells.

### 3.5. Autophagy Promotes Nephrocyte Survival but Is Dispensable for Slit Diaphragm Formation and Nephrocyte Function

Having confirmed basal autophagy in nephrocytes, we wanted to explore its role in the inhibition of autophagy. First, we studied the cell size of nephrocytes upon expression of *Atg5*-RNAi, which resulted in a significant reduction (Figure 5A,B). This effect mimics attenuation of mTOR, which may be a compensatory cellular response to lack of autophagy. Consistent with diminished mTOR activity upon inhibition of autophagy, we observed a mild reduction of the depth of labyrinthine channels by *Atg5* silencing (Figure 5C,D), but subcortical actin appeared unchanged (Figure 5E,F). In contrast, expression of the slit diaphragm protein Sns was significantly elevated based on reporter fluorescence (Figure 5G,H). Since reduced mTOR activity had no effect on Sns expression, this effect seems independent from mTOR and directly related to autophagy. 

Silencing *Atg5* further caused an increase in TUNEL-positive nephrocytes, indicating that autophagy is needed to promote nephrocyte survival (Figure 5I,J). The nephrocytes still appeared to be in the earlier stages of cell death, as live/dead staining did not indicate immediately dead cells (Appendix A). Nephrocytes are terminally differentiated cells, and autophagy may be required to sustain the cells under constant high endocytic and degradative activity. To test the functional impact of *Atg5* silencing, we performed the modified FITC-albumin assay and detected no reduction of FITC-albumin endocytosis upon disruption of autophagy (Figure 5K,L). Autophagy thus seems dispensable for nephrocyte endocytic function despite impaired survival. Consistent with the undiminished functional state, we observed a normal morphology and distancing of slit diaphragms after the knockdown of *Atg5* (Figure 5M,N). We confirmed these results using an additional RNAi directed against *Atg5* (Appendix A). Similar to podocytes, for which autophagy is dispensable without additional stressors [38], nephrocytes thus form regular slit diaphragms and remain functional without basal autophagy.

### 3.6. Inhibition of Autophagy Selectively Rescues Slit Diaphragm Misspacing Associated with Attenuation of mTOR Signaling

The increased distances between slit diaphragms that follow mTOR inhibition were not explained by expression levels of *sns* (nephrin) or cell size. However, inhibition of mTOR signaling entails excessive autophagy, which in turn might be the cause of slit diaphragm misspacing. To explore such a connection, we silenced *Atg5* concomitantly with *Tor*. To compensate for the presence of a second UAS-regulated transgene, which might dilute GAL4 and decrease the expression of *Tor*-RNAi, we compared the double knockdown of *Tor*/*Atg5* to *Tor*-RNAi expressed together with a control knockdown (*EGFP*-RNAi, Figure 6A). Indeed, the slightly weaker *GAL4* driver *Dot*-*GAL4* and the addition of a second UAS target for the control knockdown entailed a slightly milder reduction of slit diaphragm density compared to Figure 1H. However, we observed a highly significant increase in slit diaphragm density after silencing *Atg5.* The average distance between slit diaphragms returned to a level close to physiological distances (Figure 6B,C, quantitation Figure 6D, average of control cells from Figure 1H indicated by dashed line). This indicates an almost complete rescue of misspacing by *Atg5*-RNAi, showing that excessive autophagy that follows inhibition of mTOR signals underlies the wider spacing of slit diaphragms. In contrast, silencing of *Atg5* had no effect on cell size (Figure 6E,F, quantitation Figure 6G), the extent of subcortical actin (Figure 6H,I, quantitation Figure 6J), and the depth of labyrinthine channels (Figure 6K,L, quantitation Figure 6M). These results suggest that these effects are direct consequences of inhibited mTOR signaling without being connected with or mediated by excessive autophagy.

Altogether, the findings of this study explore the role of mTOR and autophagy in podocyte-like nephrocytes. Unexpectedly, the phenotypic analysis indicated that Tor-mediated excessive autophagy alters slit diaphragm spacing. Thus, our findings pave the road for further studies to elucidate the role of autophagy in the glomerular filtration barrier using the versatile *Drosophila* nephrocyte model.

## 4. Discussion

The current study investigated the role of the mTOR signaling pathway and autophagy in *Drosophila* nephrocytes. We demonstrated wider spacing between slit diaphragms in these podocyte-like cells upon mTOR inhibition, while the gain-of-function of this pathway showed regular spacing, suggesting that distances are limited by other factors. mTOR signaling regulates cell size and depth of the membrane invaginations needed for the endocytic function of nephrocytes which correlated with the extent of the subcortical actin network. Both the gain- or loss-of-function of mTOR reduced endocytic uptake in nephrocytes, suggesting that a fine balance is needed for proper nephrocyte function. Survival of nephrocytes required mTOR activity and cells showed early stages of cell death at the larval stage. Basal autophagy in nephrocytes is required for survival and limits the expression of *sns* (nephrin) but is dispensable for slit diaphragm formation and endocytic functions. Rescue experiments indicate that excessive autophagy associated with inhibition of mTOR is almost entirely responsible for slit diaphragm misspacing. Thus, we discover a direct connection between mTOR-dependent autophagy and the slit diaphragm architecture.

Here, the groundwork has been laid to establish nephrocytes as a new model to study the mechanistic role of mTOR and autophagy for podocytes. Due to the surprising molecular and functional conservation between the nephrocyte model and podocytes, similar mechanisms are quite conceivable. Importantly, this genetically tractable invertebrate model is fit to facilitate mechanistic analysis of these pathways in connection to the slit diaphragm. Supporting our model’s adequacy, we observed a significant phenotypic correlation between podocytes and nephrocytes in response to inhibition of the mTOR pathway. Genetic deletion of mTORC1 in mice entailed foot process effacement in podocytes [13], and treatment with mTOR inhibitors causes or aggravates proteinuria dose-dependently in about 10% of renal transplant recipients [42]. While we indeed found that mTOR was required for the survival and functional maintenance of the filtration barrier, autophagy seemed dispensable for basic nephrocyte function in unstressed conditions, which is in line with the observations in podocytes. As cells with exceptionally high endocytic activity, nephrocytes show an unusual pattern of the widely used reporter GFP-mCherry-Atg8a that lacks the formation of large autophagosomes, and the extent of autophagy correlates negatively with abundance of the reporter protein. The fusion protein is generated at steady rates GAL4-dependently. During autophagy, the protein tags are cleaved, and the protein is degraded in autophagolysosomes during this process. The extent of GFP-derived reporter fluorescence thus correlates with autophagic flux providing a useful and practicable read-out. Naturally, specific aspects are not conserved in evolution from fly to humans. This includes the role of mTOR signaling in the transcriptional regulation of nephrin. We did observe an induction of Sns (nephrin) expression following excessive mTOR activity (via *Tsc1*-RNAi), but this more likely reflects a reduction in autophagy since this result was equally observed with direct inhibition of autophagy. In contrast, mTOR inhibition had no effect on *sns* expression. Autophagy may exert a direct or indirect inhibitory regulation for the *sns* promoter or play a role in the trafficking of the nephrin ortholog that undergoes cycles of endocytosis and recycling. 

We observed an increased distance between slit diaphragms upon mTOR-dependent autophagy, which was almost entirely reversed by attenuation of autophagy. This supports a direct role of autophagy in the slit diaphragm architecture. However, the question remains, by what route. Since nephrocytes were smaller upon Tor inhibition, cell size cannot explain the wider spacing. With the unchanged expression of the *Drosophila* nephrin in smaller cells, the opposite outcome of denser slit diaphragms would be expected unless the protein half-life is altered. One may speculate that increased autophagy promotes the degradation of Sns (nephrin) over recycling. In any case, the wider spacing likely corresponds to a lack of slit diaphragm protein, while specific mechanisms appear to further regulate spacing between slit diaphragms. The rescue by inhibition of autophagy may be the consequence of the enhanced expression of Sns (nephrin) that was observed with inhibition of autophagy. Potentially, the reduction of slit diaphragms and shallower membrane invaginations are part of a convergent regulatory response to metabolic cues in nephrocytes. A reduction in channels that follows the reduced density of slit diaphragms will reduce the energy consumption of these cells for the extensive endocytic processes that occur incessantly within the nephrocyte’s channels. In analogy to podocytes [43] additional signaling cues may be upstream of autophagy in nephrocytes. Identification of the upstream events and the immediate target of the regulatory connection between autophagy and the spacing of slit diaphragms will require further investigation. Nevertheless, an autophagy-dependent misspacing of slit diaphragms may correspond to the first step, eventually leading to foot process effacement that was observed with mTOR attenuation in murine podocytes [13].

In conclusion, our data support the suitability of the garland cell nephrocyte model to investigate the mechanistic role of autophagy, and we present a direct connection between these pathways and the spacing between slit diaphragms.

## Figures and Tables

**Figure 1 cells-11-02103-f001:**
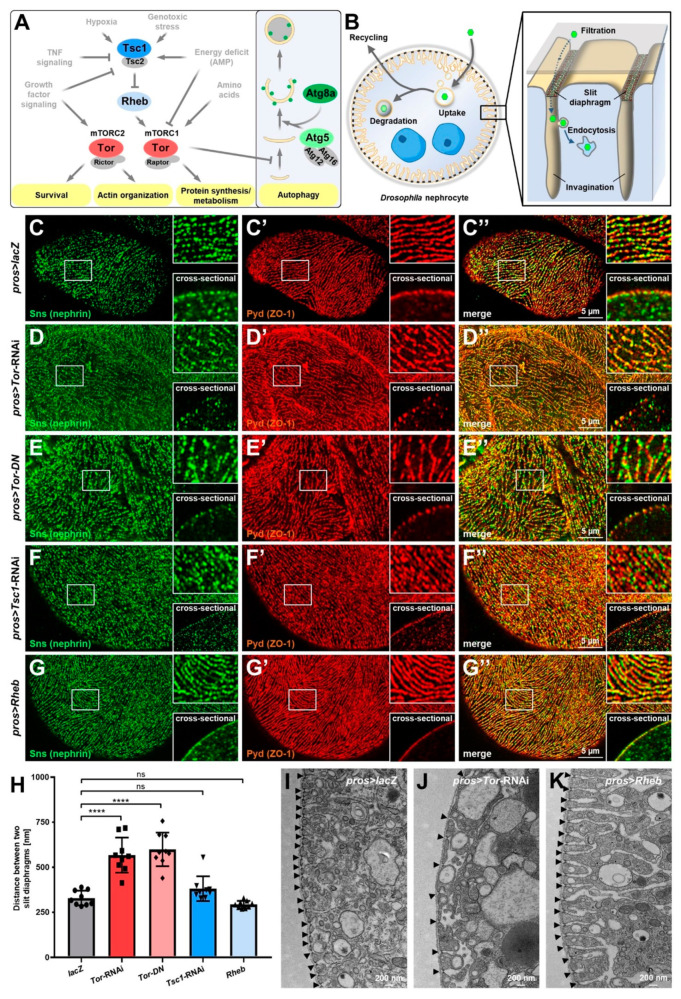
Inhibition of mTOR signaling increases spacing of slit diaphragms in *Drosophila* nephrocytes. (**A**) Simplified representation of the mTOR and autophagy pathway. The signaling pathway components targeted in this study are highlighted by colored circles. (**B**) Schematic illustration of the ultrastructure and basic function of podocyte-like *Drosophila* nephrocytes. The bi-nucleated nephrocytes remove undesired particles (green hexagons) from the larval plasma (hemolymph) by endocytic uptake for subsequent storage, degradation, or recycling. Endocytosis occurs in membrane invaginations covered by a bi-layered filtration barrier consisting of the basement membrane (gray surface) and the slit diaphragm formed by the proteins Sns and Kirre (depicted in green and red). (**C**–**G’’**) Confocal microscopy of nephrocytes co-stained for Sns (nephrin) and Pyd (ZO-1). Magnified regions of the tangential sections are shown in the upper insets, and surface details from cross-sections in the lower insets. *lacZ* expressing control cells display the regular fingerprint pattern of slit diaphragms (**C**–**C’’**) while cells expressing either *Tor*-RNAi (**D**–**D’’**) or a dominant-negative variant of *Tor* (*Tor-DN*, **E**–**E’’**) under control of *pros*-*GAL4* show wider spacing. In contrast, induction of mTOR signaling by expression of *Tsc1*-RNAi (**F**–**F’’**) or overexpression of *Rheb* (**G**–**G’’**) results in a regular slit diaphragm pattern comparable to control cells. (**H**) Quantification of the distance between two slit diaphragms is shown analogous to conditions in (**C**–**G’’**). Distances were measured along a linear path representing the widest diameter of individual cells. Data shows mean ± standard deviation, *n* = 9 animals per genotype with three cells for each animal (every dot, square or triangle represents one animal of the indicated genotype). Statistical differences were assessed by one-way ANOVA with post-hoc analysis, *p* < 0.0001 (****) for *Tor*-RNAi, *p* < 0.0001 (****) for *Tor-DN*, *p* > 0.05 (ns) for *Tsc1*-RNAi, *p* > 0.05 (ns) for *Rheb*. (**I**–**K**) Transmission electron microscopy (TEM) images of cells expressing *lacZ* (**I**), *Tor*-RNAi (**J**), or overexpressing *Rheb* (**K**). Increased and irregular distances of slit diaphragms are exclusively observed upon expression of *Tor*-RNAi (slit diaphragms highlighted by arrowheads). Inhibition of mTOR further reduced the depth of the labyrinthine channels representing membrane invaginations. In contrast, labyrinthine channels are deeper when mTOR signaling is induced by overexpression of *Rheb* while slit diaphragm distances are regular.

**Figure 2 cells-11-02103-f002:**
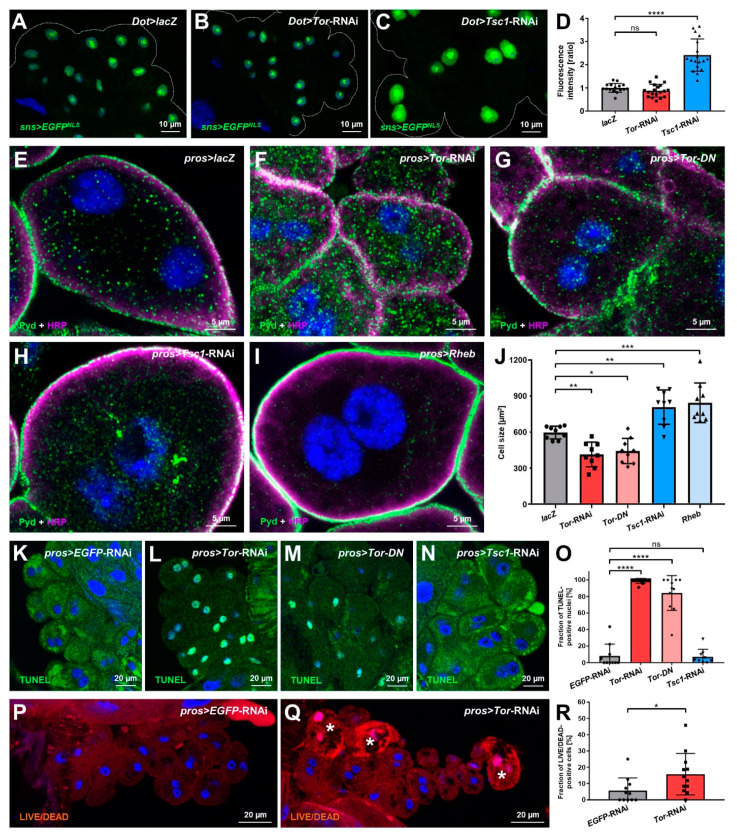
Manipulations of the mTOR pathway alter nephrocyte cell size, survival, and *sns* expression. (**A**–**C**) Expression of *EGFP* with a nuclear localization signal (EGFP^NLS^) under control of a 2 kb enhancer region of the *Drosophila sns* promoter reflects endogenous *sns* expression levels in nephrocytes. The expression of *Tor*-RNAi (**B**) under control of *Dot-GAL4* does not alter EGFP^NLS^ intensity compared to control cells expressing *lacZ* (**A**). Conversely, the expression of *Tsc1*-RNAi (**C**) leads to increased EGFP^NLS^ levels. Nuclei are stained by Hoechst 33342 in blue here and throughout the figure. (**D**) Quantification of nuclear EGFP-derived fluorescence intensity is normalized to the average of *lacZ* expressing controls and shown for the indicated genotypes (**A**–**C**). Data shows mean intensity ± standard deviation as an average of the three brightest representative cells from one animal, *n* = 16–19 animals per genotype (every dot, square or triangle represents one animal of the indicated genotype here and throughout the figure). Statistical differences were assessed by one-way ANOVA with post-hoc analysis, *p* > 0.05 (ns) for *Tor*-RNAi, *p* < 0.0001 (****) for *Tsc1*-RNAi. (**E**–**I**) Nephrocytes were stained for Pyd to assess cell size. Co-staining with HRP confirms the correct localization of Pyd at the cell membrane. Cell size is smaller in cells expressing either *Tor*-RNAi (**F**) or a dominant-negative variant of *Tor* (*Tor-DN*, (**G**)) compared to the *lacZ* expressing control cells (**E**) under the control of *pros-GAL4*. In contrast, cells expressing *Tsc1*-RNAi (**H**) or overexpressing *Rheb* (**I**) are enlarged compared to control cells. (**J**) Quantification of cell size by measurement of the area outlined by Pyd staining in the cross-sectional plane in conditions analogous to (E–I). Data shows mean ± standard deviation, *n* = 9 animals per genotype with three cells for each animal. Statistical differences were assessed by one-way ANOVA with post-hoc analysis, *p* < 0.01 (**) for *Tor*-RNAi, *p* < 0.05 (*) for *Tor-DN*, *p* < 0.01 (**) for *Tsc1*-RNAi, *p* < 0.001 (***) for *Rheb*. (**K**–**N**) Fragmented DNA, which associates with cell death, is reflected by TUNEL staining. Cells expressing either *Tor*-RNAi (**L**) or a dominant-negative variant of *Tor* (*Tor-DN*, (**M**)) show highly elevated levels of TUNEL-positive nuclei compared to cells expressing a control RNAi (**K**) or *Tsc1*-RNAi (**N**). (**O**) Quantification of the fraction of TUNEL-positive nuclei in conditions analogous to (K–N). Data shows mean ± standard deviation, *n* = 9–11 animals per genotype with 18–20 cells on average for each animal. Statistical differences were assessed by one-way ANOVA with post-hoc analysis, *p* < 0.0001 (****) for *Tor*-RNAi, *p* < 0.0001 (****) for *Tor-DN*, *p* > 0.05 (ns) for *Tsc1*-RNAi. (**P**,**Q**) The LIVE/DEAD fixable dye is excluded from viable cells, but in dead cells with an impaired membrane barrier, the dye accumulates. This results in a brighter fluorescence, in particular, within the nucleus. Increased numbers of dead cells (highlighted by stars) are detectable among cells expressing *Tor*-RNAi (**Q**) compared to cells expressing a control RNAi (**P**). (**R**) Quantification of the fraction of LIVE/DEAD-positive cells in conditions analogous to (P–Q). Data shows mean ± standard deviation, *n* = 11–12 animals per genotype with 19–23 cells on average for each animal. Statistical difference was assessed by unpaired *t*-test, *p* < 0.05 (*) for *Tor*-RNAi.

**Figure 3 cells-11-02103-f003:**
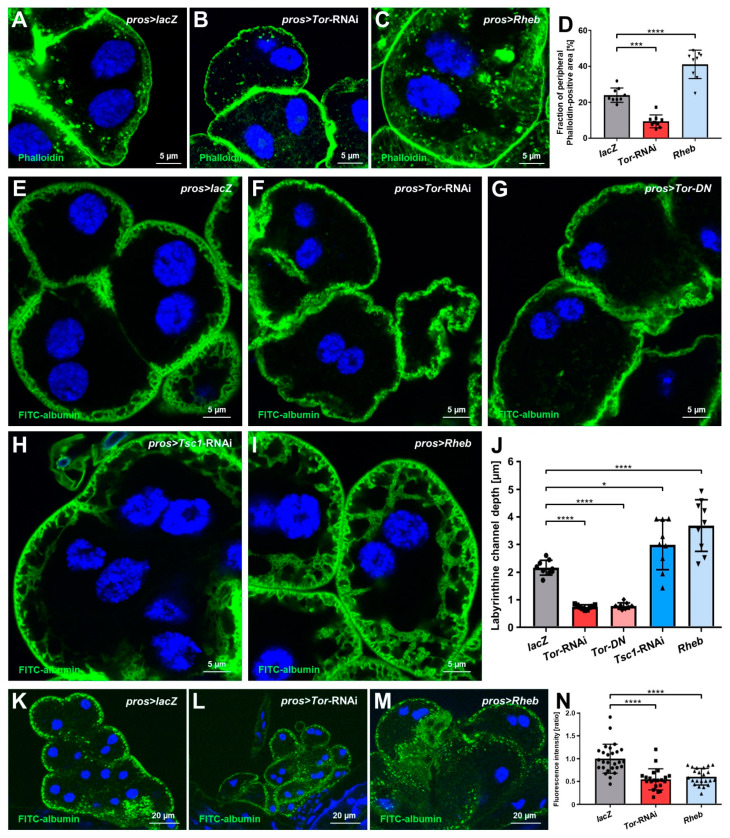
The depth of peripheral actin network and labyrinthine channels corresponds to mTOR activity, while nephrocyte function decreases upon both mTOR gain- or loss-of-function. (**A**–**C**) Actin marker phalloidin shows a regular peripheral actin network in *lacZ* expressing control cells (**A**) with linear cortical actin and the adjoining subcortical actin network. Upon expression of *Tor*-RNAi (**B**) under the control of *pros-GAL4*, the actin network loses depth but becomes deeper with overexpression of *Rheb* (**C**). Nuclei are stained by Hoechst 33342 in blue here and throughout the figure. (**D**) Quantification of the fraction of the cellular area in the cross-sectional plane that is covered by the peripheral actin network for the conditions analogous to (**A**–**C**). Data shows mean ± standard deviation, *n* = 9 animals per genotype with three cells for each animal (every dot, square or triangle represents one animal of the indicated genotype here and throughout the figure). Statistical differences were assessed by one-way ANOVA with post-hoc analysis, *p* < 0.001 (***) for *Tor*-RNAi, *p* < 0.0001 (****) for *Rheb*. (**E**–**I**) Nephrocyte labyrinthine channels are visualized by passive FITC-albumin tracer diffusion into the channels after brief fixation. Channels are shortened in cells expressing either *Tor*-RNAi (**F**) or a dominant-negative variant of *Tor* (*Tor-DN*, **G**) compared to the *lacZ* expressing control cells (**E**). Conversely, cells expressing *Tsc1*-RNAi (**H**) or overexpressing *Rheb* (**I**) show deeper channels compared to control cells. (**J**) Quantification of labyrinthine channel depth analogous to conditions in (**E**–**I**) based on three representative measurements in sections of the cell surface that are not in immediate proximity to a neighboring cell. Data shows mean ± standard deviation, *n* = 9 animals per genotype with three cells for each animal. Statistical differences were assessed using a one-way ANOVA with post-hoc analysis, *p* < 0.0001 (****) for *Tor*-RNAi, *p* < 0.0001 (****) for *Tor-DN*, *p* < 0.05 (*) for *Tsc1*-RNAi, and *p* < 0.0001 (****) for *Rheb*. (**K**–**M**) Nephrocyte function is visualized by the FITC-albumin endocytosis assay. Interestingly, both the expression of *Tor*-RNAi (**L**) and overexpression of *Rheb* (**M**) result in reduced tracer uptake compared to control cells (**K**). (**N**) Quantification of FITC-albumin-derived fluorescence intensity is normalized to the average of a *lacZ* expressing control experiment performed in parallel and shown for the indicated genotypes (**K**–**M**). Data shows mean ± standard deviation as the average of the three brightest representative cells from one animal, *n* = 21–27 animals per genotype. Statistical differences were assessed by one-way ANOVA with post-hoc analysis, *p* < 0.0001 (****) for *Tor*-RNAi, and *p* < 0.0001 (****) for *Rheb*.

**Figure 4 cells-11-02103-f004:**
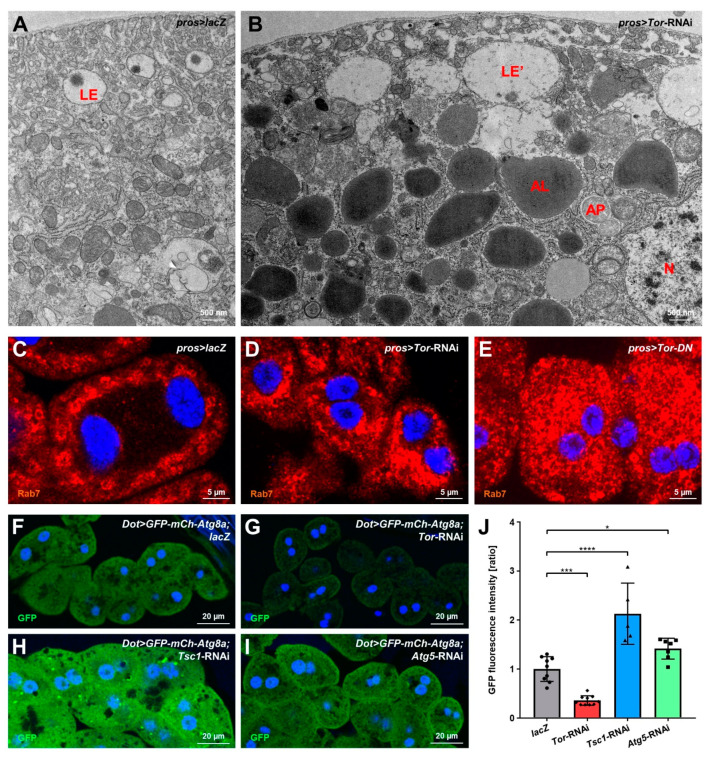
mTOR inhibition in nephrocytes enhances autophagy. (**A**,**B**) Transmission electron microscopy (TEM) images of cells expressing *lacZ* (**A**), or *Tor*-RNAi (**B**). Upon expression of *Tor*-RNAi under the control of *pros-GAL4*, autophagosomes, autophagolysosomes, and various other autophagic vesicles become more numerous. AL: autophagolysosome; AP: autophagosome; LE: late endosome; LE’: aberrant late endosome; N: nucleus. (**C**–**E**) Nephrocytes were stained for Rab7 to visualize the late endosomal compartment. Expression of *Tor*-RNAi (**D**) or a dominant-negative variant of *Tor* (*Tor-DN*, **E**) results in a strong expansion and accumulation of Rab7-positive vesicles compared to *lacZ* expressing control cells (**C**). Nuclei are stained by Hoechst 33342 in blue here and throughout the figure. (**F**–**I**) Fluorescence-tagged Atg8a labels various stages of autophagy, thereby serving as a marker for autophagic activity. Co-expression of tagged *Atg8a* and *Tor*-RNAi (**G**) under control of *Dot-GAL4* results in decreased GFP-derived fluorescence intensity compared to control cells expressing *lacZ* (**F**). In contrast, induction of mTOR signaling by *Tsc1*-RNAi (**H**) and inhibition of autophagic activity by *Atg5*-RNAi (**I**) leads to increased fluorescence intensity. Fluorescence intensity negatively correlates with autophagic activity. (**J**) Quantification of cellular fluorescence intensity is normalized to the average of *lacZ* expressing controls and shown for the indicated genotypes (**F**–**I**). Data shows mean ± standard deviation as an average of the three brightest representative cells from one animal, *n* = 5-10 animals per genotype (every dot, square or triangle represents one animal of the indicated genotype). Statistical differences were assessed by one-way ANOVA with post-hoc analysis, *p* < 0.001 (***) for *Tor*-RNAi, *p* < 0.0001 (****) for *Tsc1*-RNAi, *p* < 0.05 (*) for *Atg5*-RNAi.

**Figure 5 cells-11-02103-f005:**
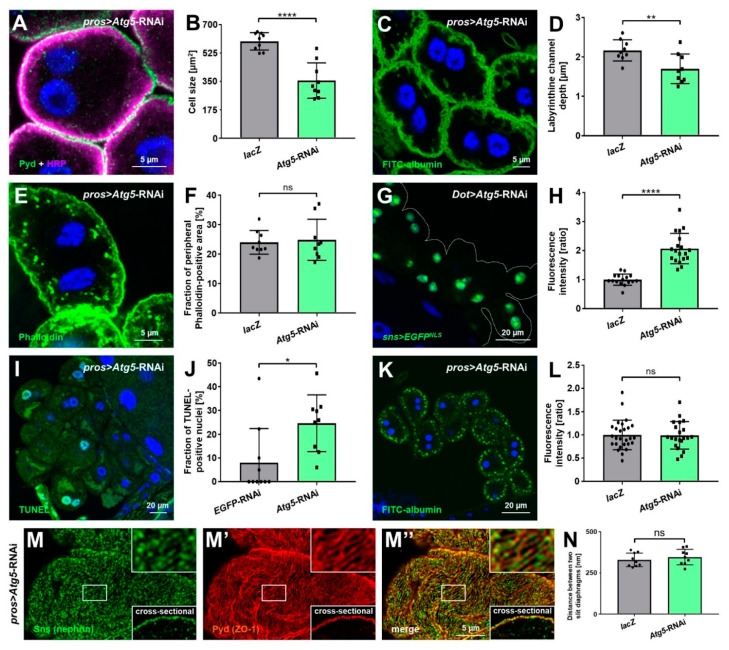
Basal autophagy in nephrocytes limits expression of *sns* (nephrin) and promotes survival but is dispensable for slit diaphragm formation and endocytic function. (**A**) Nephrocytes were stained for Pyd to assess cell size. Co-staining with HRP confirms the correct localization of Pyd at the cell membrane. Cell size is smaller in cells expressing *Atg5*-RNAi under the control of *pros-GAL4* compared to the *lacZ* expressing control cells (Figure 2E). (**B**) Quantification of cell size by measurement of the area outlined by Pyd staining in the cross-sectional plane in conditions analogous to (A and Figure 2E). Data shows mean ± standard deviation, *n* = 9 animals per genotype with three cells for each animal (every dot and square represents one animal here and throughout the figure). Statistical difference was assessed by unpaired *t*-test, *p* < 0.0001 (****) for *Atg5*-RNAi. (**C**) Nephrocyte labyrinthine channels are visualized by passive FITC-albumin tracer diffusion into the channels after brief fixation. Channels are shortened in cells expressing *Atg5*-RNAi compared to the *lacZ* expressing control cells (Figure 3E). (**D**) Quantification of labyrinthine channel depth analogous to conditions in ((**C**) and Figure 3E) based on three representative measurements in sections of the cell surface that are not in immediate proximity to a neighboring cell. Data shows mean ± standard deviation, *n* = 9 animals per genotype with three cells for each animal. Statistical difference was assessed by unpaired *t*-test, *p* < 0.01 (**) for *Atg5*-RNAi. (**E**) Actin marker phalloidin shows a regular peripheral actin network in *lacZ* expressing control cells (Figure 3A) with linear cortical actin and the adjoining subcortical actin network. Expression of *Atg5*-RNAi does not alter the peripheral actin network. (**F**) The quantification of the fraction of the cellular area in the cross-sectional plane that is covered by the peripheral actin network for the conditions analogous to ((**E**) and Figure 3A). Data shows mean ± standard deviation, *n* = 9 animals per genotype with three cells for each animal. Statistical difference was assessed by unpaired *t*-test, *p* > 0.05 (ns) for *Atg5*-RNAi. (**G**) The expression of *EGFP* with a nuclear localization signal (EGFP^NLS^) under control of a 2 kb enhancer region of the *Drosophila sns* promoter reflects endogenous *sns* expression levels in nephrocytes. The expression of *Atg5*-RNAi leads to increased EGFP^NLS^ levels compared to control cells expressing *lacZ* (Figure 2A). (**H**) The quantification of nuclear EGFP-derived fluorescence intensity is normalized to the average of *lacZ* expressing controls and shown for the indicated genotypes (G and Figure 2A). Data shows mean ± standard deviation as an average of the three brightest representative cells from one animal, *n* = 16–19 animals per genotype. Statistical difference was assessed by unpaired *t*-test, *p* < 0.0001 (****) for *Atg5*-RNAi. (**I**) Fragmented DNA, which associates with cell death, is reflected by TUNEL staining. Cells expressing *Atg5*-RNAi show elevated levels of TUNEL-positive nuclei compared to cells expressing a control RNAi (Figure 2K). (**J**) Quantification of the fraction of TUNEL-positive nuclei in conditions analogous to ((**I**) and Figure 2K). Data shows mean ± standard deviation, *n* = 9–10 animals per genotype with 18–23 cells on average for each animal. Statistical difference was assessed by unpaired *t*-test, *p* < 0.05 (*) for *Atg5*-RNAi. (**K**) Nephrocyte function is visualized by the FITC-albumin endocytosis assay. The expression of *Atg5*-RNAi does not alter tracer uptake compared to control cells (Figure 3K). (**L**) Quantification of FITC-albumin-derived fluorescence intensity is normalized to the average of a *lacZ* expressing control experiment performed in parallel and shown for the indicated genotypes ((**K**) and Figure 3K). Data shows mean ± standard deviation as an average of the three brightest representative cells from one animal, *n* = 21–27 animals per genotype. Statistical difference was assessed by unpaired *t*-test, *p* > 0.05 (ns) for *Atg5*-RNAi. (**M**–**M’’**) Nephrocytes were co-stained for Sns (nephrin) and Pyd (ZO-1). Magnified regions of the tangential section are shown in the upper insets, and surface details from cross-sections in the lower insets. Cells expressing *Atg5*-RNAi display the regular slit diaphragm fingerprint pattern of control cells expressing *lacZ* (Figure 1C–C’’). (**N**) Quantification of the distance between two slit diaphragms is shown analogous to conditions in (M–M’’ and Figure 1C–C’’). Distances were measured along a linear path representing the widest diameter of individual cells. Data shows mean ± standard deviation, *n* = 9 animals per genotype with three cells for each animal. Statistical difference was assessed by unpaired *t*-test, *p* > 0.05 (ns) for *Atg5*-RNAi.

**Figure 6 cells-11-02103-f006:**
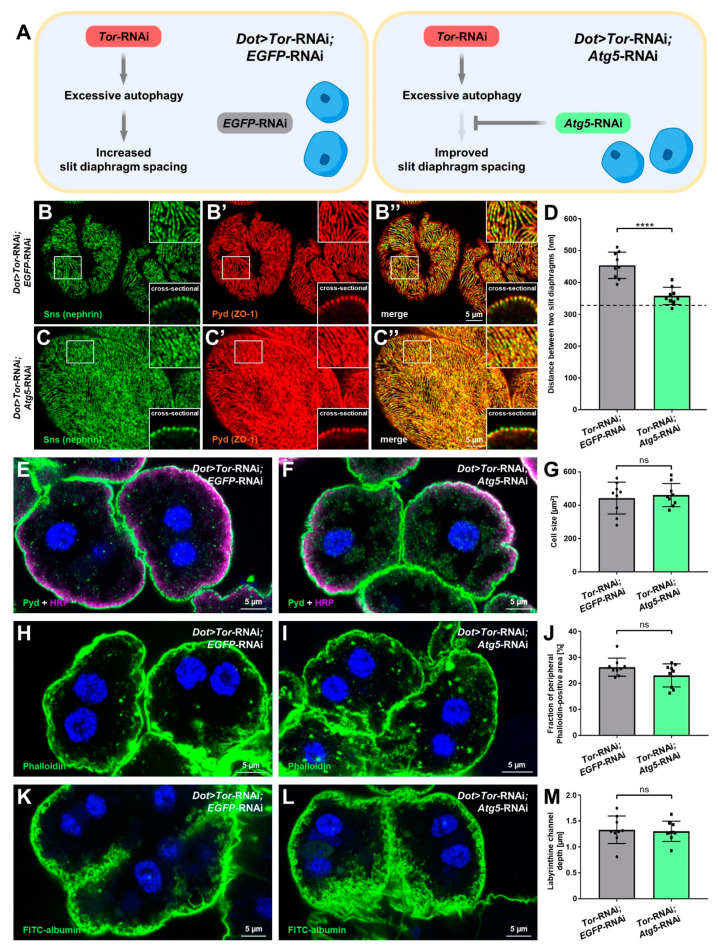
Inhibition of autophagy selectively rescues slit diaphragm misspacing associated with attenuation of mTOR signaling. (**A**) The schematic illustrates the genetic rescue experiment. To test the role of excessive autophagy for slit diaphragm spacing, *Atg5* is silenced together with *Tor* (left). To control for the effect of the presence of two UAS targets, a control knockdown (*EGFP*-RNAi) is combined with *Tor* silencing (right). (**B**–**C’’**) Nephrocytes were co-stained for Sns (nephrin) and Pyd (ZO-1). Magnified regions of the tangential section are shown in the upper insets, and surface details from cross-sections in the lower insets. Co-expression of *Atg5*-RNAi in nephrocytes expressing *Tor*-RNAi (**C**–**C’’**) under control of *Dot-GAL4* partially rescues the widened slit diaphragm distance that results from the co-expression of *Tor*-RNAi and a control RNAi (**B**–**B’’**). (**D**) Quantification of the distance between two slit diaphragms is shown analogous to conditions in (**B**–**C**’’). The dashed line indicates the average slit diaphragm distance of control cells (Figure 1C). Data shows mean ± standard deviation, *n* = 9 animals per genotype with 3 cells each animal (every dot and square represents one animal of the indicated genotype here and throughout the figure). Statistical difference was assessed by unpaired *t*-test, *p* < 0.0001 (****) for *Tor*-RNAi;*Atg5*-RNAi. (**E**,**F**) Nephrocytes were stained for Pyd to assess cell size. Co-staining with HRP confirms correct localization of Pyd at the cell membrane. Cell size is unchanged in cells co-expressing *Atg5*-RNAi and *Tor*-RNAi (**F**) compared to the *EGFP*-RNAi;*Tor*-RNAi expressing cells (**E**). Nuclei are stained by Hoechst 33342 in blue here and throughout the figure. (**G**) Quantification of cell size by measurement of the area outlined by Pyd staining in the cross-sectional plane in conditions analogous to (**E**,**F**). Data shows mean ± standard deviation, *n* = 9 animals per genotype with three cells for each animal. Statistical difference was assessed by unpaired *t*-test, *p* > 0.05 (ns) for *Atg5*-RNAi. (**H**,**I**) Actin marker phalloidin shows the actin network. The extent of the peripheral actin network is unchanged in cells co-expressing *Atg5*-RNAi and *Tor*-RNAi (**I**) compared to the *EGFP*-RNAi;*Tor*-RNAi expressing cells (**H**). (**J**) Quantification of the fraction of the cellular area in the cross-sectional plane that is covered by the peripheral actin network for the conditions analogous to (**H**–**I**). Data shows mean ± standard deviation, *n* = 9 animals per genotype with three cells for each animal. Statistical difference was assessed by unpaired *t*-test, *p* > 0.05 (ns) for *Atg5*-RNAi. (**K**,**L**) Nephrocyte labyrinthine channels are visualized by passive FITC-albumin tracer diffusion into the channels after brief fixation. Channel depth is unchanged in cells expressing both *Atg5*-RNAi and *Tor*-RNAi (**L**) compared to the *EGFP*-RNAi; *Tor*-RNAi expressing cells (**K**). (**M**) Quantification of labyrinthine channel depth analogous to conditions in (**K**,**L**). Data shows mean ± standard deviation, *n* = 9 animals per genotype with three cells for each animal. Statistical difference was assessed by unpaired *t*-test, *p* > 0.05 (ns) for *Atg5*-RNAi.

## Data Availability

Not applicable.

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
