# Peer review of "mTOR-Dependent Autophagy Regulates Slit Diaphragm Density in Podocyte-like Drosophila Nephrocytes"

_cells, 2022, doi:10.3390/cells11132103_

Round 1

Author Response

Response Reviewer 1:

Therefore, the mTOR data, which is at the focus of this work, is very convincing and I only have minor comments for this part. In comparison to the mTOR data, the data on autophagy is weaker: only one RNAi line and no confirmation of the impact of Atg5-RNAi on the activity of autophagy. Therefore, if the authors want to conclude that autophagy is “dispensable for slit diaphragm formation and basic nephrocyte function”, further experimental support is needed. Specifically, the author need to confirm the impact of Atg5-RNAi on autophagic activity or at least, clearly state these limitations.

Response:

Thank you very much for reviewing the manuscript and the many specific and helpful comments. We agree with the reviewer about this limitation and we expanded our analysis following the reviewer’s suggestion now including an additional RNAi line directed against Atg5, which returned quite comparable results to the first Atg5-RNAi. The additional experimental data is now included in the manuscript as a new Supplemental Figure:

We confirmed these results using an additional RNAi directed against Atg5 (Supplemental Figure 5E-M)

Overall, this is a well-written paper with thoroughly designed and well-executed experiments. As the study investigates an interesting and relevant question on podocyte biology, I am confident that it will be of high interest to the nephrology research community.

Response:

Thank you very much!

Specific comments.

- The authors investigate the role of mTOR on the subcortical actin network, which reflects the labyrinth channels and on the depth of the labyrinthine channels by labeling. They found that mTOR inhibition decreased the subcortical actin network and the depth of the labyrinth channels, while mTOr activation had the opposite effect.

à On first glance, the TEM images in Fig. 1 I-K seem to confirm the observations on the depth of the labyrinth channels. Maybe the authors could evaluate the images under this aspect?

Response:

Thank you for this suggestion. We quantified the depth of labyrinthine channels in our electron microscopy images, which confirmed our findings using confocal microscopy. Original images and the quantification are now included in the manuscript as a new Supplemental Figure 3.

- TEM analysis upon inhibition of mTOR shows an increased amount of endocytic vesicles, in particular structures that the authors identify as late endosomes and autophago-lysosomes. This matches the increased signal of Rab7 observed in mTOR-deficient nephrocytes. They then use overexpressed Atg8a as a marker for autophagosomes and find that atg8a is rapidly degraded in mTOR deficient cells, while an inhibition of autophagy by hyperactive mTOR or Atg5 knockdown increases the Atg8a level. This matches the depicted TEM image, where only few autophagosomes are seen. à For better characterization of the accumulating vesicles, the authors should perform immunostaining for other marker proteins, such as e.g. LAMP2 or LC3, and quantify the fluorescence intensity in mTOR deficient nephrocytes.

Response:

We completely understand the reviewer’s concern about possible limitations of an analysis based on electron microscopy. However, time constraints prevented us from performing a more comprehensive analysis since obtaining and establishing antibodies would not be possible within the allocated time for the revision. On the other hand, we would like to point out that “transmission electron microscopy (EM) is the classical method of visualizing and clearly distinguishing autophagic vesicles” (PMID: 28704946). We toned down the manuscript text to be more precise:

Vesicles with the morphology of autophagosomes were rare in control nephrocytes (Figure 4A), but became quite abundant in cells expressing Tor-RNAi and vesicles representing various stages of autophagy were present (Figure 4B). The most abundant vesicles appeared to be electron-dense autophagolysosomes, which is consistent with a strong expansion of Rab7-positive vesicles in Tor-RNAi or dominant negative Tor compared to control (Figure 4C-E). To rule out lipid droplets that also show high electron-density, we performed Nile Red staining, but this lipid dye seemed much less abundant than autophagosomes (Supplemental Figure 4). Thus, our morphologic analysis in TEM strongly suggests excessive autophagy in nephrocytes upon Tor inhibition.

- In the next part of the study, the authors focus on analyzing the impact of direct inhibition of autophagy and conclude that “autophagy promotes nephrocyte survival but is dispensable for slit diaphragm formation and nephrocyte function”.

à To support the conclusion that autophagy is dispensible for nephrocyte and slit diaphragm function, the data should be strengthened by confirmatory experiments in a second fly line for autophagy inhibition and it should be verified to which extend autophagy is inhibited in Atg5-deficient nephrocytes.

Response:

It was not possible to perform the combination crosses for a rescue using the second Atg-RNAi as well. However, since the findings from both RNAis matched nicely and the rescue is a highly specific outcome, we feel that the risk of this being an off-target effect seems negligible.

à The inhibition of autophagy as well as the inhibition of mTOR (activation of autophagy) activate apoptosis in nephrocytes. For mTOR inhibition, the authors also showed that the number of dead cells as validated by a live/dead dye was increased (but to a lesser extent than TUNEL positivity). I wonder whether autophagy inhibition also increases the number of dead cells or whether the effect is weaker than the one of mTOR inhibition. Interestingly, the inhibition of autophagy resulting from mTOR inhibition (Tcs1-RNAi) appears to be insufficient to activate apoptosis in nephrocytes.

Response:

The effect of autophagy inhibition appears weaker than mTOR inhibition. Accordingly, we detected a much lower number of TUNEL positive cells. Following the reviewer’s suggestion, we now also performed the live/dead assay, where we detected no significant increase. The additional data was included in the manuscript:

Inhibition of autophagy had a weaker impact on nephrocyte survival than mTOR inhibition and nephrocytes still appeared to be in earlier stages of cell death, since live/dead staining did not indicate immediately dead cells (Supplemental Figure 5A-D).

- In the final part of the part, genetic rescue experiments in mTOR-RNAi nephrocytes show that the increased distances between neighbouring SDs are improved upon inhibition of autophagy. This suggests that hyperactive autophagy is a major contributor to the observed slit diaphragm defects. As this is a very interesting and major finding of this paper, it would be nice to have a second validation with e.g. the TOR-DN flies.

Response:

It was not possible to perform the combination crosses for testing Tor-DN in this manner within the time assigned for the review. However, since the findings of Tor-RNAi and Tor-DN matched nicely in a specific manner, the risk of an off-target effect seems extremely low.

Reviewer 2 Report

The authors have highlighted the use of the drosophila podocyte model for mechanistic studies on the role of mTOR signaling and autophagy and found a direct mTOR/autophagy-dependent regulation of the slit diaphragm architecture. The manuscript is well-written. However, the authors need to consider the following points to improvise the manuscript.

1. Describe the electron microscopy method in detail. Which instrument was used (Company, model)?

2. It is unclear in the manuscript whether the authors are discussing mTORC1 or mTORC2 in the results section?

3. What happens to the downstream signaling pathway of mTOR?

Author Response

Response Reviewer: 2

Comments and Suggestions for Authors

The authors have highlighted the use of the drosophila podocyte model for mechanistic studies on the role of mTOR signaling and autophagy and found a direct mTOR/autophagy-dependent regulation of the slit diaphragm architecture. The manuscript is well-written. However, the authors need to consider the following points to improvise the manuscript.

Response:

Thank you very much for reviewing the manuscript and the encouraging comments!

  1. Describe the electron microscopy method in detail. Which instrument was used (Company, model)?

Response:

We apologize for this omission, this information has now been included in the materials and methods section: The TEM imaging was performed with a Thermo Scientific Talos L120C TEM.

It is unclear in the manuscript whether the authors are discussing mTORC1 or mTORC2 in the results section?

Response:

We are happy to clarify this point. Since Tor is a critical part of both complexes, our manipulation using RNAi and a dominant negative version is expected to affect both, mTORC1 and mTORC2 (see also Figure 1A). Since both are affected in the setting we used here, we made no distinction between both complexes.

  1. What happens to the downstream signaling pathway of mTOR?

Response:

We are happy to clarify that survival, actin organization, autophagy and transcriptional regulation are all part of the effects triggered downstream of mTOR (Figure 1A). We feel that our analysis (mainly in Figure 2 and Figure 3) establishes that signaling downstream of Tor has strong effects on nephrocyte function, morphology, cytoskeleton, survival and fly nephrin expression which must be mediated downstream of Tor.

Round 2

Reviewer 1 Report

The authors have satisfactorily addressed my concerns and have responded to them appropriately. I feel that in its current form the manuscript is suited for publication. I have no additional comments.